# Loxin Reduced the Inflammatory Response in the Liver and the Aortic Fatty Streak Formation in Mice Fed with a High-Fat Diet

**DOI:** 10.3390/ijms23137329

**Published:** 2022-06-30

**Authors:** Camila Reyes, Estefanía Nova-Lamperti, Daniel Duran-Sandoval, Daniela Rojas, Jorge Gajardo, Enrique Guzman-Gutierrez, Camila Bustos-Ruiz, Valeska Ormazábal, Felipe A. Zúñiga, Carlos Escudero, Claudio Aguayo

**Affiliations:** 1Department of Clinical Biochemistry and Immunology, Faculty of Pharmacy, University of Concepcion, Concepción 4030000, Chile; camireyesviver@gmail.com (C.R.); enova@udec.cl (E.N.-L.); dduran@udec.cl (D.D.-S.); eguzman@udec.cl (E.G.-G.); camilbustos@udec.cl (C.B.-R.); fzuniga@udec.cl (F.A.Z.); 2Department of Animal Pathology, Faculty of Veterinary Sciences, University of Concepcion, Chillan 3787000, Chile; drojasm@udec.cl; 3Department of Internal Medicine, Faculty of Medicine, University of Concepcion, Concepción 4030000, Chile; jgncardio@gmail.com; 4Department of Pharmacology, Faculty of Biological Sciences, Universidad de Concepción, Concepción 4030000, Chile; vormazabal@udec.cl; 5Vascular Physiology Laboratory, Department of Basic Sciences, Universidad del Bio-Bio, Chillan 3787000, Chile; cescudero@ubiobio.cl; 6Group of Research and Innovation in Vascular Health (GRIVAS Health), Chillan 3787000, Chile

**Keywords:** LOXIN, Lox-1, vascular dysfunctions, liver disease

## Abstract

Oxidized low-density lipoprotein (ox-LDL) is the most harmful form of cholesterol associated with vascular atherosclerosis and hepatic injury, mainly due to inflammatory cell infiltration and subsequent severe tissue injury. Lox-1 is the central ox-LDL receptor expressed in endothelial and immune cells, its activation regulating inflammatory cytokines and chemotactic factor secretion. Recently, a Lox-1 truncated protein isoform lacking the ox-LDL binding domain named LOXIN has been described. We have previously shown that LOXIN overexpression blocked Lox-1-mediated ox-LDL internalization in human endothelial progenitor cells in vitro. However, the functional role of LOXIN in targeting inflammation or tissue injury in vivo remains unknown. In this study, we investigate whether LOXIN modulated the expression of Lox-1 and reduced the inflammatory response in a high-fat-diet mice model. Results indicate that human LOXIN blocks Lox-1 mediated uptake of ox-LDL in H4-II-E-C3 cells. Furthermore, in vivo experiments showed that overexpression of LOXIN reduced both fatty streak lesions in the aorta and inflammation and fibrosis in the liver. These findings were associated with the down-regulation of Lox-1 in endothelial cells. Then, LOXIN prevents hepatic and aortic tissue damage in vivo associated with reduced Lox-1 expression in endothelial cells. We encourage future research to understand better the underlying molecular mechanisms and potential therapeutic use of LOXIN.

## 1. Introduction

Atherosclerosis is an inflammatory condition characterized by lipid metabolic alteration and a maladaptive inflammatory response. Moreover, the accumulation of free cholesterol leads to the activation of the inflammasome and cytokines, inflammation, and fibrosis in the liver [1,2,3]. Indeed, high-fat diet-fed mice (HFD) have shown aggravated hepatic steatosis, resulting in inflammatory cell infiltration and subsequent severe hepatic injury [4].

In humans, increased levels of oxidized low-density lipoprotein (ox-LDL) have been associated with a high incidence of metabolic syndrome, acute coronary events, hepatocellular injury, and fibrosis mediated by lectin-like oxidized LDL receptor 1 (Lox-1) [4,5]. Therefore, Lox-1 has been studied in pathological conditions, such as atherosclerosis, diabetes, coronary artery disease, and portal venous inflammation and fibrosis in the liver [6,7,8]. Thus, it has been demonstrated that ox-LDL upregulates TNF-α expression in rat Kupffer cells through a Lox-1-dependent pathway [7,8,9]. In addition, Lox-1 overexpression promotes vascular dysfunction and atherosclerotic and liver diseases; thus, targeting or blocking Lox-1 is a potential therapeutic strategy to reduce the detrimental consequence of ox-LDL uptake. [4,9].

In this context, alternative splicing of the *OLR1* gene (Lox-1) leads to the expression of an exon five-truncated protein lacking the lectin binding domain called LOXIN [10,11,12]. Studies in peripheral blood macrophages have shown that increased expression of LOXIN inhibited ox-LDL-induced apoptosis [13,14]. Similarly, experiments using co-transfection of LOXIN and Lox-1 in HEK293 cells, human endothelial progenitor cells (hEPC), and ECV304 cells have revealed that LOXIN exerted a protective effect against ox-LDL-induced apoptosis [15,16,17]. Therefore, LOXIN has a pro-survival role in cells exposed to ox-LDL. However, it is unknown whether LOXIN overexpression plays a protective role in metabolic syndrome development by reducing liver inflammation and atherosclerosis in a high-fat-diet (HFD) mouse model.

## 2. Result

### 2.1. Ox-LDL and Lysophosphatidylcholine (LPC) Reduced Viability and Modulated Lox-1 Expression in H4-II-E-C3 Cells

Previous data from our research group have identified the detrimental effect of ox-LDL on inducing apoptosis in endothelial cells; however, it was unclear whether this damage was extended to other tissues, such as the liver. Thus, the viability of H4-II-E-C3 cells (rat hepatoma cell line) after incubation with increasing concentrations of ox-LDL or LPC for 24 h was evaluated. Results showed that ox-LDL reduced H4-II-E-C3 cell viability when ox-LDL concentration was above 100 mg/mL (Figure 1A). Similar results were observed in H4-II-E-C3 cells when LPC was used above 50 mg/mL (Figure 1B). Then, we evaluated whether these stimuli modulate the expression of Lox-1 (Figure 1C). Thus, incubation with ox-LDL at 50 μg/mL induced a 2.5-fold increase in Lox-1 expression, while no effect was observed in higher concentrations of ox-LDL. Similarly, the use of 1 μg/mL but not higher concentrations (50 and 100 μg/mL) of LPC increased the protein expression of Lox-1 (Figure 1C).

### 2.2. Overexpression of Ado-LOXIN in H4-II-E-C3 Cells Reduced the Internalization of Ox-LDL

Previous studies have shown the in vitro protective effects of human LOXIN in human-derived endothelial cells [17]. Thus, we evaluate the same approach in our in vitro model with H4-II-E-C3 cells. Since the C-type lectin (CTL)/C-type lectin-like (CTLD) domain from the *OLR1* gene (Lox-1) is highly conserved, we evaluated the effect of human LOXIN in the rat hepatocyte cell line. Increasing concentrations of a vector pAdTrack-LOXIN coupled with Green Fluorescent Protein (GFP) were used to induce LOXIN overexpression in H4-II-E-C3 cells, evidenced by the GFP signal. Our results revealed a maximum expression of GFP-LOXIN above 200 CFU/mL (Figure 1D).

After defining the concentration of ox-LDL and GFP-LOXIN, the effect of LOXIN expression over ox-LDL internalization was evaluated by measuring Apo B-100, the major structural component of LDL stained with Alexa Fluor 592 (red) in the rat (H4-II-E-C3) and human (DU145) cell lines expressing GFP-LOXIN by immunofluorescence. DU145 was used as a positive control as this human cell line has a constitutive expression of Lox-1. The GFP-LOXIN-expressing cells in both cell lines did not internalize Apo B100 (red), suggesting that LOXIN overexpression reduced ox-LDL internalization (data no show). These results confirm previous evidence published by our group suggesting that the incorporation of oxLDL was abolished entirely in the endothelial progenitor cell pre-treated with 100 CFU/cell of vector Ado-Loxin [17].

### 2.3. LOXIN Overexpression Reduced Lipid Deposition in Aortas

We then characterize our HFD mouse model by measuring several parameters such as fasting glucose levels, lipid profile, alanine aminotransferase (ALT), and aspartate aminotransferase (AST), and tissue structures, 12 weeks post-dietary intervention. Compared to ND mice, HFD mice exhibited increased glucose, total cholesterol (TC), ALT, and AST levels (Figure 2, Appendix A). Then, we analyzed the effect of LOXIN expression on these variables, and the data revealed that LOXIN expression reduced transaminases. However, it does not change the circulating levels of cholesterol in comparison with the ND mice (Appendix A, Figure 2). In addition, no significant changes were observed in glucose levels, lipid profile, ALT, and AST in animals fed a normal diet and treated with Ado-LOXIN.

In terms of lipid deposition in aortic plaques and vessel intima disorganization, we observed a higher deposit of lipids in the aortas (Figure 3A) and more disorganization (Figure 3B) in tissues from HFD animals than their respective counterpart in the ND group, confirming the metabolic-induced impairments by the HFD.

Lipid deposition and vessel intima disorganization were also compared between ND and HFD mice injected with Ado-LOXIN or Ad-Null. However, only histological analysis indicates that HFD increases the thickness of the aorta. Interestingly, Ado-LOXIN expression appears to recover this abnormality (Figure 4A).

Finally, immunohistochemical analyses were then performed to determine the presence of Lox-1 in the aortas (Figure 4B). Our results showed staining for Lox-1 in the aortas, specifically in the vessel wall (endothelial cells), in the HFD group. When the effect of LOXIN on Lox-1 expression was evaluated in the aortas of HFD mice, we observed that Ado-LOXIN infection reduced staining for Lox-1 compared to mice infected with Ad-Null (Figure 4B), suggesting a decrease in Lox-1 expression.

### 2.4. Effect of LOXIN Overexpression on Hepatic Tissues

Since our model’s transaminase circulating levels were reduced by LOXIN expression, we analyze hepatic inflammation in the HFD model. Inflammatory infiltration was found only in the liver of HFD mice compared to ND (Figure 5A). Microscopic analysis showed that these infiltrations were mainly inflammatory cells. Liver samples from HFD mice exhibited fatty infiltration, lipid inclusions, and accumulations, preferentially in the lymphocytes and polymorphs. In this group, we also observed steatosis, associated with some distortion in the hepatic architecture, compared to ND mice. Mice treated with Ado-LOXIN in the HFD group show improvement in the hepatic accumulation of lipid, steatosis, hepatocyte ballooning, and fibrosis compared with the HFD group and Ado-null (Figure 5A). Consistently, the non-alcoholic fatty liver disease (NAFLD) activity score (NAS) (for calculating the grade of steatosis, inflammation, and ballooning) in HFD mice increased (score two steatoses, two inflammation, and two ballooning) and was significantly lower in HFD mice infected with the Ado-LOXIN vector (score 0 steatoses, 0 inflammation, and one ballooning). The NAS scores in ND mice compared to ND mice infected with the Ado-LOXIN vector do not differ significantly (Figure 5A).

Second-harmonic generation (SHG) was used to determine fibrillar collagen and hepatocyte morphology changes during liver fibrosis progression. Figure 5B shows changes in collagen deposit by Sirius Red staining (insert shows collagen deposit in grey color). In ND livers, hepatocytes are healthy and low collagen is distributed. HFD results in high collagen distribution that Ado-LOXIN blocks. Thus, quantification of SHG images shows that HFD induces a 15-fold increase in fluorescence (0.2 *v/s* 3.16 mm^2^ collagen area/total area), which is significantly inhibited by Ado-LOXIN (1.3 mm^2^ collagen area/total area). No differences were observed in the fluorescence intensity under ND and treatment with ado-LOXIN (0.29 mm^2^ collagen area/total area). At the same time, infiltration of inflammatory cells was not different between the ND groups with or without adenovirus. Ado-LOXIN significantly enhances collagen amount in the ND mice, suggesting hepatic damage. However, quantification of hepatic enzymes was not significantly elevated compared with ND mice (Appendix A).

Moreover, we performed immunostaining of Lox-1 in liver sections to confirm the contribution of the receptor in the hepatic injury process (Figure 6). Hepatocytes from ND groups exhibited low and diffuse staining of Lox-1 in the tissue. In contrast, liver from HFD mice showed an increased expression of Lox-1, mainly located on endothelial cells, hepatocytes, and perivascular inflammatory cells. Interestingly, the HFD group infected with the Ado-LOXIN vector changed the pattern of expression of Lox-1. Thus, there was a decrease in the location of Lox-1 in vascular cells, while no changes in the expression in hepatocytes were found (Figure 6).

We also examined inflammation in the liver by evaluating immunostaining for both tumor necrosis factor-alpha (TNF-α) and interleukin 6 (IL-6) (Figure 7 and Figure 8). The hepatic expression of these two proinflammatory markers was higher in HFD mice when compared with ND mice. Notably, infection with Ado-LOXIN decreased the pattern of expression of TNF-α (Figure 7) and IL-6 (Figure 8) expression in liver parenchyma of HFD mice compared with ND mice and Ado-null. Also, no changes in cytokine expression were found in the perivascular region. In contrast, diffuse expression was detected in control mice given adenovirus null.

## 3. Discussion

Lox-1 has been recognized as one of the main receptors that mediates oxLDL uptake, promoting several changes such as foam cell formation, endothelial dysfunction, apoptosis, and inflammation. In this context, LOXIN, a truncated form of Lox-1 that cannot internalize oxLDL, is a potential candidate with the ability to block Lox-1-mediated responses. Our present study found that LOXIN overexpression inhibited the internalization of ox-LDL in hepatoma cells in vitro. In addition, HFD mice exhibited higher glucose levels and active inflammation in the liver (ALT and AST), which were reverted in the presence of LOXIN. Moreover, LOXIN decreased the aortic fatty streak formation, reduced Lox-1 expression in hepatic endothelial cells, and reduced the inflammation in mice fed with a high-fat diet. Altogether, the results indicate that LOXIN overexpression prevents the development and progression of atherosclerosis and favors liver tissue repair.

Lox-1 is a specific cell-surface receptor for ox-LDL expressed in endothelial cells, vascular smooth muscle cells, and adipocytes. Lox-1 activated by oxLDL leads to endothelial dysfunction, a critical pathophysiological step in the atherosclerotic process. Also, this receptor is active during inflammatory processes, such as chronic liver disease. Accordingly, we observed that Lox-1 expression is upregulated after stimulation with ox-LDL and LPC in a hepatic cell line (H4-II-E-C3). Therefore, our results further confirmed the inhibition of LOXIN capacity of human and mouse cells. In this manuscript, we further extend these findings to analyze the potential use of Ado-LOXIN to protect mouse cells from Lox-1 activation. However, this upregulation was observed only at low concentrations in the dose–response curve for both stimuli. Previous studies have reported the expression of Lox-1 under physiological conditions in murine Kupffer cells [18,19]; and human hepatic sinusoidal endothelial cells [18]. Also, Lox-1 was detected in circulating macrophages under pathological conditions, such as hepatic disease [20,21]. Therefore, we confirm previously published data regarding the expression of Lox-1 in liver tissue. Also, previous findings indicate that Lox-1 is activated by ox-LDL and promotes proinflammatory mechanisms.

In terms of cytotoxicity induced by ox-LDL or LPC, our results showed that both stimuli were cytotoxic at high concentrations in H4-II-E-C3 cells. Similarly, previous studies have demonstrated ox-LDL-mediated cytotoxicity in other cancer cell lines, including HT29 (colon), OVCAR3 (ovarian), MCF-7 (lung), and PC (prostate) cells [22,23,24]. In addition, we reported that Lox-1 stimulation by ox-LDL (50 μg/mL) induced apoptosis in the vascular cell, and human umbilical endothelial cells and endothelial progenitor cells [17].

To address whether the human LOXIN affects rat Lox-1-mediated ox-LDL internalization, an internalization assay of Apo B100 was carried out in hepatoma cells [12]. In the dimer formation (lox-1/lox1) is essential for the interaction of ox-LDL with Lox-1 [25,26,27,28]. Whether the human LOXIN protein can interact with and inhibit the mouse Lox-1 protein is unknown. Our experiments using human LOXIN overexpressed in rat hepatocytes suggest that the latter can block the mouse Lox-1 protein. Similarly, Biocca et al. observed reduced ox-LDL internalization when COS cells were co-transfected with LOXIN and human Lox-1 [13,27]. Despite these observations, the mechanism behind the interaction between rat Lox-1 and human LOXIN remains unclear. It has been suggested that Lox-1 and LOXIN may interact in the secretory compartment, especially in the endoplasmic reticulum and Golgi apparatus, where they continue their maturation until reaching the cell surface [28]. Furthermore, Biocca et al. have shown that heterodimerization between wild-type Lox-1 and a dominant-negative Lox-1 receptor reduced the expression of wild-type Lox-1 and the uptake of ox-LDL [27]. These results support the idea that LOXIN can block Lox-1 in human and mice cells.

Infection of adenoviral vectors has been used to investigate the function of any given protein in vivo [29,30,31].

The study in vivo includes the mouse strains C57BL/6. The C57BL/6 mouse is more susceptible to develop early atherosclerosis among the inbred mouse strains. The lesions induced after feeding a high-fat diet are early fatty streaks containing principally cholesterol-rich macrophages. However, an important issue in this study is the cholic acid supplementation. Cholic acid has been shown to possess proinflammatory properties, which may have an accelerating influence on atherosclerosis progression. These results confirm previous studies which have shown that at this stage, C57BL/6 mice develop significant fatty streak lesions at the aortic root [32,33]. Paigen et al. examined the location and timing of lesion formation when mice were fed with the same atherogenic diet as used in the present study and they found that foam cells were present at the aortic root of C57BL/6 mice 6 weeks after initiation of the diet.

On the other hand, our result revealed decreased lipid accumulation when HFD mice were treated with the ado-LOXIN. This observation was associated with decreased Lox-1 protein abundance observed in the immunohistochemistry analysis of the aorta. These results agree with previous reports showing a decrease in atherosclerotic plaque formation in Apo E-knockout mice overexpressing human Lox-1 and LOXIN [34]. Other studies have confirmed the use of adenoviral vectors in treating some pathologies. For example, adenovirus overexpressing IL-10 (Ad.IL-10), the administration of Ad.IL-10 resulted in extended systemic expression of IL-10 peak serum level (3.0 ± 1.1 ng/mL) and reduced atherosclerotic lumen stenosis by 62.2% [34,35], showing that local IL-10 administration is a promising novel therapy in the treatment of developing atherosclerotic lesions at defined vascular sites. Other groups have evidenced that adenoviral vectors that can be administered systemically can achieve the desired effect of the protein when injected into a mouse model of cavernous nerve injury or breast cancer [36,37]. Song et al. (2014) demonstrated the effectiveness of adenoviruses encoding Smad7 gene (Ad-Smad7) on erectile function in a mouse model of cavernous nerve injury. Its result shows that the adenovirus restored erectile function by enhancing endothelial cell function [37]. Hu et al. 2010 demonstrate the administration of an adenoviral vector expressing the soluble form of TGFβ receptor II fused with human Fc IgG1 (sTGFβRIIFc) gene intravenously into MDA-MB-231 human xenograft-bearing mice, resulting in significant inhibition of tumour growth and production of sTGFβRIIFc in the blood [36].

These findings suggest that inhibition of the adenovirus vector may represent a promising therapeutic strategy for cancer and atherogenesis therapy. Therefore, overexpression of LOXIN via reduction of Lox-1 expression and reduction of ox-LDL internalization may constitute a potential therapy in atherosclerosis. Importantly, other publications support our results showing that systemic administration of adenoviral vectors can effectively target diseased livers. These authors suggest that the minor changes observed by the use of adenoviruses are associated with the minimum dose used (1 × 10^8^ vp/kg) since it was shown that high doses (4 × 10^12^ vp/kg) resulted in increased mortality (100% in cirrhotic animals and approximately 60% in normal rats). A 1.2 × 10^13^ vp/kg dose killed all cirrhotic animals and 80% healthy animals, whereas 1.2 × 10^12^ vp/kg was reasonably tolerated in normal and cirrhotic rats with an optimal degree of efficiency of liver transduction [37,38,39,40].

As described previously, increased Lox-1 activity in endothelial cells and macrophages may promote the progression of atherosclerosis (AS) [2]. However, Lox-1 is also found on the surface of liver cells, suggesting a promising protective role provided by hepatic Lox-1 expression. In the liver, Lox-1 is expressed in activated macrophages in the proinflammatory environment [6,7], in particular on the surface of liver cells [20,21,41], and has been involved in portal venous plaque inflammation [21,42], or sinusoidal endothelial dysfunction generated by activated hepatic Kupffer and stellate cells [41]. A similar result shows that Lox-1 is involved in liver microcirculation disturbance and nonalcoholic fatty liver disease (NAFLD) [43,44,45]. Dorman et al. (2006) and Li et al. (2011) suggest that ox-LDL can activate hepatic Kupffer and stellate cells, trigger inflammation and fibrogenesis, and induce sinusoidal endothelial dysfunction [44,46].

Our results show that HFD induced liver damage, an effect inhibited by ado-LOXIN associated with a significant reduction of endothelial cell Lox-1 expression in the liver. Similar evidence by Zhang et al., 2014, shows that oxLDL significantly increased Lox-1 expression at mRNA and protein levels in human liver sinusoidal endothelial cells (HLSECs). oxLDL stimulation increased ROS generation and NFκB activation, upregulated ET1 and caveolin 1 expression, downregulated eNOS expression, and reduced the fenestra diameter and porosity. All these oxLDL-mediated effects were inhibited after Lox-1 knockdown [42]. In addition, Ado-LOXIN significantly enhances collagen amount without changes in the hepatic enzymes in the ND mice. Further analysis needs to clarify whether LOXIN overexpression itself has any harmful effect on the liver.

Associated with the changes in the expression of Lox-1, we observed an increase in the levels of TNF-α and IL-6, two of the most widely studied pro-inflammatory cytokines released from inflammatory cells in the liver. Clinical and experimental data demonstrated that expression of the TNF-α and IL-6 genes are increased and positively correlated with the severity of the liver disease in human and animal models [41,42]. TNF-α has been found to induce hepatic inflammation and subsequent apoptosis associated with insulin resistance [37]. While IL-6 induces and accelerates mononuclear cell accumulation in inflammatory environments. Nevertheless, TNF-α and IL-6 are also critical in the liver regeneration process, acting as pro-mitogens [46,47,48]. Furthermore, antibodies against Lox-1 neutralized its pro-inflammatory signaling mediated by TNF-α and IL-6 [49,50].

We did not analyze the potential involvement of these cytokines in the hepatic response. However, our results showed that TNF-α and IL-6 are increased in animals treated with Ado-LOXIN, suggesting that the protective effect of LOXIN depends on the release of these cytokines in the liver parenchyma.

One of our study’s weaknesses is knowing the cellular mechanism involvement of inhibition of LOXIN on LOX-1. We are trying to overcome this limitation through the inclusion of cellular models. Several studies show that oxLDL binding to Lox-1 stimulates downstream signaling, including MAPK, protein kinase C, octamer-binding protein-1, and PI3K/Akt [51,52]. We have evidence in endothelial cells that the inhibition of p38MAPK is necessary to protect cells of free radicals produced by oxLDL. However, the mechanism responsible for the repair process induced by LOXIN in vivo remains established.

## 4. Materials and Methods

### 4.1. LDL Isolation and Ox-LDL Preparation

Native lipoproteins were obtained from the human plasma of normolipidemic volunteers. The Ethics Committee approved the experimental protocol of the Faculty of Pharmacy of Universidad de Concepción, and all volunteers signed informed consent. Plasma samples were withdrawn using Ethylenediaminetetraacetic acid tetrasodium (EDTA, 0.5 mg/mL) (Sigma-Aldrich, St Louis, MO, USA) as an anticoagulant. LDLs were isolated by sequential potassium bromide density centrifugation. After removing chylomicrons, very-low-density lipoprotein, and intermediate-density lipoprotein, the native LDLs were yielded at a final density of 1.019 to 1.063 g/mL [53].

Ox-LDL was obtained by exposing native LDL to CuSO4 (10 μM) for 4 h at 37 °C, getting a delta of absorbance at 234 nm of 0.4 units, as previously reported [53]. The oxidation process was ended with EDTA (2 mM) and Butylhydroxytoluene (BHT, 4.5 μM) (Sigma-Aldrich, St Louis, MO, USA). Ox-LDL was dialyzed with PBS, and LDL-protein concentration was quantified with a Protein DC kit (BioRad Laboratory, Hertfordshire, UK). Lysophosphatidylcholine (LPC, present in up to 40% of the total lipid content of ox-LDL) (Sigma-Aldrich, St Louis, MO, USA) was used in a parallel experiment measure its effect on Lox-1 in H4-II-E-C3 cells.

### 4.2. H4-II-E-C3 and COS Culture

Rat hepatoma cell line H4-II-E-C3 and COS cells (ATCC, Manassas, VA, USA) were grown in Dulbecco’s Modified Eagle’s Medium (DMEM; Lonza, Walkersville, MD, USA) supplemented with 10% fetal bovine serum and 2% Glutamax (Invitrogen, Carlsbad, CA, USA) in an incubator at 37 °C with 5% CO_2_. Cells were seeded at a density of 5 × 10^5^ cells/well (12-well plates) in a complete culture medium.

### 4.3. Western Blotting

Total protein was obtained from confluent 6 well dishes. Cells were washed twice with ice-cold phosphate buffer solution (PBS) and harvested in 150 μL of lysis buffer (63.7 mM Tris/HCl (pH 6.8); 10% glycerol, 2% sodium dodecylsulphate; 1 mM sodium orthovanadate. Cells were sonicated for 10 min in an ice/water bath, and total protein was separated by centrifugation (10,000× *g*, 15 min, 4 °C). Then, proteins (50 µg) were separated by polyacrylamide gel (10%) electrophoresis and transferred to polyvinylidene difluoride membranes (Thermo Scientific. Waltham, MA, USA). The proteins were probed with primary polyclonal goat anti-Lox-1 (1/1000, 12 h, 4 °C) (R&D Systems, Minneapolis, MN, USA), and polyclonal rabbit anti-β-actin (1/2000, 12 h, 4 °C) (Thermo Scientific. Waltham, MA, USA). Membranes were rinsed in Tris buffer saline-tween 20 (TBS-T/0.1%) and incubated for 1 h in TBS-T containing 2.5% milk and secondary horseradish peroxidase-conjugated goat anti-goat or anti-rabbit antibodies (1/2000) (Abcam, Cambridge, UK). Finally, proteins were detected by chemiluminescence reagent (GE Healthcare, Piscataway, NJ, USA) and quantified by densitometry using Image J 1.47v plugging.

### 4.4. Generation of the Adenoviral Vector for the Overexpression of the Protein LOXIN

The molecule was designed based on the nucleotidic sequence of the Homo Sapiens LDL receptor-1 (*OLR1*) transcript variant 3 mRNA (NM-001172633.1). The sequence of LOXIN was extracted by enzymatic digestion using *XhoI* and *SalI* (New England Biolabs, Ipswich, MA, USA), obtaining a fragment of 582 bp. The band was purified from agarose gel 1% by the Silica Bead DNA Gel Extraction Kit (Thermo Scientific. Waltham, MA, USA). It was inserted by enzymatic ligation to the shuttle-vector pAdTrack-CMV (Quantum Biotechnologies, Montreal, Canada), previously digested with *XhoI* and *SalI*. The recombinant shuttle vector was linearized with EcoRI, while DNA was purified from 1% agarose gel with QIAquick Gel Extraction Kit (QIAGEN, Crawley, UK). The insertion of the gene of interest into the adenoviral genome was obtained by homolog recombination in *E. coli* BJ5183 by co-electroporation with the supercoiled form of the plasmid pAd-Easy-1 (Quantum Biotechnologies, Montreal, Canada) in a proportion of 1 μg of the linearized vector and 100 ng of pAd-Easy-1. The restriction analysis identified the positive recombinants with *SalI* and *PacI* (New England Biolabs, Ipswich, USA). The recombinants were transformed in *E. coli* Top10 (Invitrogen, Carlsbad, CA, USA), and the plasmid was purified by the NucleoBond^®^ Xtra Midi kit (Macharey-Nagel, Allentown, PA, USA) [17].

The transfection of the recombinant plasmid obtained the adenoviral particles before digestion with the enzyme PacI (New England Biolabs, Ipswich, MA, USA) using the satisfactions transfection reagent (Agilent Technologies, Santa Clara, CA, USA). The transfection was performed in a 100 mm plate with HEK-293 at 70% confluence using 15 μg of the digested plasmidial DNA following manufacturer instructions. At 15 days later, infective virions were observed as the lysis halos around the infected cells. Serial dilutions performed the viral titration in a 96-well culture dish with HEK-293 cells at a 70–80% of confluence. It was defined as a viral particle or colony-forming unit (CFU) to the highest dilution at which at least one fluorescent cell is visualized per well. The result is expressed in colony-forming units per ml (CFU/mL).

The multiplicity of infection (MOI) was determined for each type of cell used (see below). In brief, 1 × 10^5^ cells were cultured in DMEM 10% fetal bovine serum (FBS) for 24 h. When cell culture reached a confluence of 80%, a dose-response concentration of viral particles (0, 5, 25, 50, 100, and 150 viral particles per cell) for 6 h was performed. The culture medium was changed for DMEM to 10% and incubated for 48 h at 37 °C with 5% CO_2_. The optimal quantity of viral particles was determined by microscopy using the relationship between the expression of the tracer protein (green fluorescent protein, GFP) and the cytopathic effect produced. A virus without insert (pAd-null) was generated similarly, and it was considered a control in the experiments [17].

### 4.5. Overexpression of LOXIN in H4-II-E-C3

H4-II-E-C3 cells (70–80% confluent) were plated in a 6-well culture dish with Dulbecco’s Modified Eagle’s Medium (DMEM; Lonza, Walkersville, MD, USA) supplemented with 10% FBS (Invitrogen, Carlsbad, CA, USA). Then, 0, 5, 25, 50, 100, 150, and 200 viral particles per cell for 6 h were added. The culture medium was then changed for DMEM 5% and incubated for 48 h at 37 °C at 5% CO_2_. Cells were lysed, and western blot followed the previously described protocol using the Lox-1 antibody (R&D Systems, Minneapolis, MN, USA).

### 4.6. Ethidium Homodimer-1 (EthD-1) Incorporation Assay for Adenoviral Transduced H4-II-E-C3 cells

H4-II-E-C3 cells were cultured for 14 days, and adenoviral transduction was performed by adding 100 viral particles per cell for 6 h. The culture medium was then changed for DMEM 5% and incubated for 48 h at 37 °C at 5% CO_2_. Subsequently, cells were treated with ox-LDL (0–200 μg/mL, 24 h) or lysophosphatidylcholine (LPC, 0–120 μg/mL 24 h), and the incorporation of Ethidium homodimer-1 (EthD-1) was performed as previously described [17].

### 4.7. Metabolic Analysis

The Hitachi biochemical instrument determined blood lipid, including serum levels of total cholesterol and triglyceride. Also, blood glucose was determined using the AccuTrend (Roche Biochemicals, GmbH, Mannheim, Germany) described by the manufacturers. Alanine aminotransferase (ALT) and aspartate aminotransferase (AST) were examined using an enzymatic assay kit (Randox, Crumlin, UK).

### 4.8. Tissue Harvesting and Quantification of Atherosclerosis

The experiment protocol was approved by the Ethics Committee of the Faculty of Pharmacy of Universidad de Concepción, Chile, and conformed to the Guide for the Care and Use of Laboratory Animal from Agencia Nacional de Investigacion y Desarrollo (ANID). The 3R principle was used in all experimental approaches using mice.

Twenty C57BL6 mice were divided into two diet conditions: mice fed with a normal diet (ND, *n* = 10), and mice fed with a high-fat diet (HFD, *n* = 10) ad libitum. The HFD contained cholic acid (0.25%), high-fat (35.0%), protein (23.0%) and carbohydrate (35.5%). Fat, protein, and carbohydrate provided were 58.0%, 16.4%, and 25.5%. Control mice received water and control chow diet ad libitum for the 13th week with calories provided by fat (4%), protein (23%), and carbohydrate (25%).

Each group was divided into two infection conditions: 5 mice were injected with an empty Ado-Null and Ado-LOXIN vectors (3 × 10^8^ CFU/mL). Infection was generated via tail injection once a week. In the 13th week of the experiment, the animals were sacrificed under euthanasia by intraperitoneal injection of pentobarbital sodium. Tissues such as the aorta (thoracic-abdominal) and liver were excised and further preserved in 4% paraformaldehyde. To analyze the lesion, the sections were stained with hematoxylin and eosin (H&E) and Oil Red O. Face total aortas were obtained from mice in study groups, and lipid deposition was quantified from 5 sections (8 mm thickness), separated by 80 mm from each other, using computer images analysis (NisElements). The intima–media thickness (IMT) was quantified from H&E preparations (15 measurements from each animal).

### 4.9. Microscopic Analysis and Immunohistochemistry

Microscopic analysis and immunohistochemistry analysis were performed as previously described [17,54]. Briefly, paraffin-embedded tissue sections were cut into 5 μm slices to use for immunodetection of Lox-1 (Abcam, Cambridge, UK) (dilution 1:500 *v/v*), IL-6 (dilution 1:500 *v/v*) and TNF-alpha (dilution 1:500 *v/v*) (Vector Laboratories, Burlingame, CA, USA). The slides were deparaffinized and hydrated, and then endogenous peroxidase was blocked in 3% hydrogen peroxide. The antigen was unmasking in 1 mM EDTA, pH 8.0 for 15 min. The antigen–antibody reaction was visualized using Immprees Universal Reagent (Abcam, Cambridge, UK) and Impact substrate DAB (Abcam, Cambridge, UK). Densitometry was performed using Image-Pro Plus software (Media Cybernetics, Rockville, MD, USA) in two random pictures from each preparation [54]. Estimation of Lox-1 levels was performed using a scale of crosses.

The non-alcoholic fatty liver disease (NAFLD) activity score (NAS) was calculated from the grade of steatosis, inflammation, and ballooning. In brief, steatosis was quantified as 0 (<5%), 1 (5–33%), 2 (>33–66%), and 3 (>66%) based on the percentage of hepatocytes containing fat droplets. Ballooning degeneration was scored as 0 (none), 1 (few), or 2 (many) according to the amount of ballooning hepatocytes. Lobular inflammation was scored as 0 (no foci), 1 (<2 foci), 2 (2–4 foci), and 3 (>4 foci) according to the inflammation foci per 200× field [55].

### 4.10. Second-Harmonic Generation (SHG) Microscopy

Second Harmonic Generation was used to visualize the collagen fibers; the samples were analyzed with a pulsed laser in a confocal microscope LSM7005. Images were acquired and treated with the built-in processing software. Briefly, the 4 μm samples from FFPE were dewaxed, hydrated, and incubated in 1% PBS-Tween-202 for 20 min at room temperature [56].

### 4.11. Statistical Analysis

Data were analyzed using standard statistical software (SPSS version 25) and GraphPad Prism 9.0. Values were expressed as mean ± SEM. Statistical data analysis was performed with a one-way ANOVA followed by a Tukey–Kramer. Differences were considered significant when the *p*-value was <0.05.

## 5. Conclusions

In conclusion, LOXIN reduced inflammatory response in the liver and the aortic fatty streak formation in mice fed with the HFD, an effect associated with regulating the hepatic levels of TNF-α and IL-6.

## Figures and Tables

**Figure 1 ijms-23-07329-f001:**
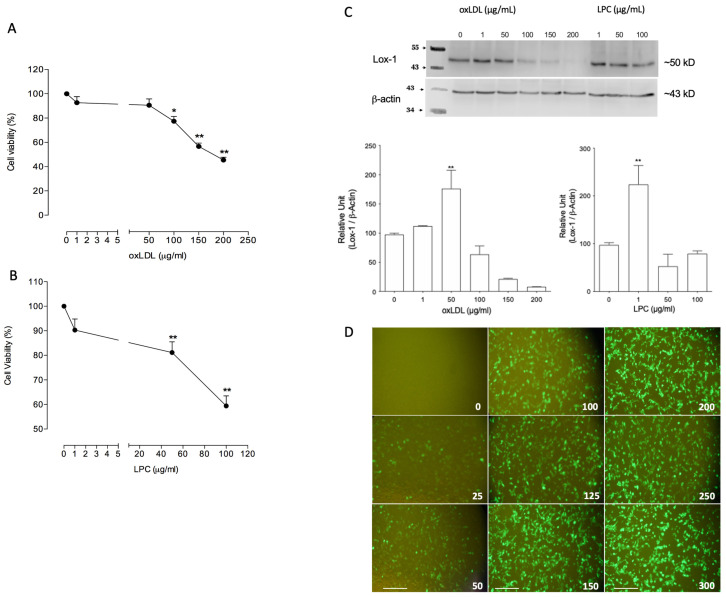
Ox-LDL and LPC induce Lox-1 expression and cytotoxicity in H4-II-E-C3 cells. Determination of cytotoxicity by Oxidized low-density lipoprotein, (ox-LDL) (**A**) or Lysophosphatidylcholine (LPC) (**B**) by incorporating ethidium homodimer-1 in H4-II-E-C3 cells. (**C**) Western blot was carried out with a specific antibody for checking the Lox-1 protein in H4-II-E-C3 cells after treatment with LPC and ox-LDL. β-Actin was used as a loading control for normalization. (**D**) The representative motive of infection (MOI) for the adenoviral vectors Ado-LOXIN in H4-II-E-C3 cells. Values are expressed as mean and SEM (*n* = 4). Scale bars indicate 200 µm. * *p* < 0.05 and ** *p* < 0.01 versus to control cells.

**Figure 2 ijms-23-07329-f002:**
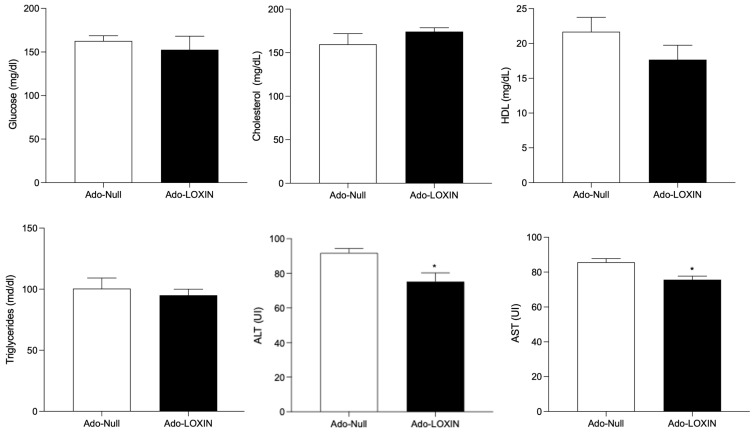
Metabolic characterization of mice fed a high-fat diet. Blood glucose, cholesterol, Triglyceride, HDL AST, and ALT were measured in mice fed a normal diet (ND) and high-fat diet (HFD) mice. Mice were infected with an adenovirus vector (1 × 10^8^ Plaque-forming unit, pfu) expressing LOXIN (Ado-LOXIN) or empty vector (Ado-Null) via the tail vein. One-way ANOVA followed by a Tukey–Kramer test was used to examine the difference between the experimental group. Values are expressed as mean, and SEM *n* = 5. Statistical significance is represented as * *p* < 0.05.

**Figure 3 ijms-23-07329-f003:**
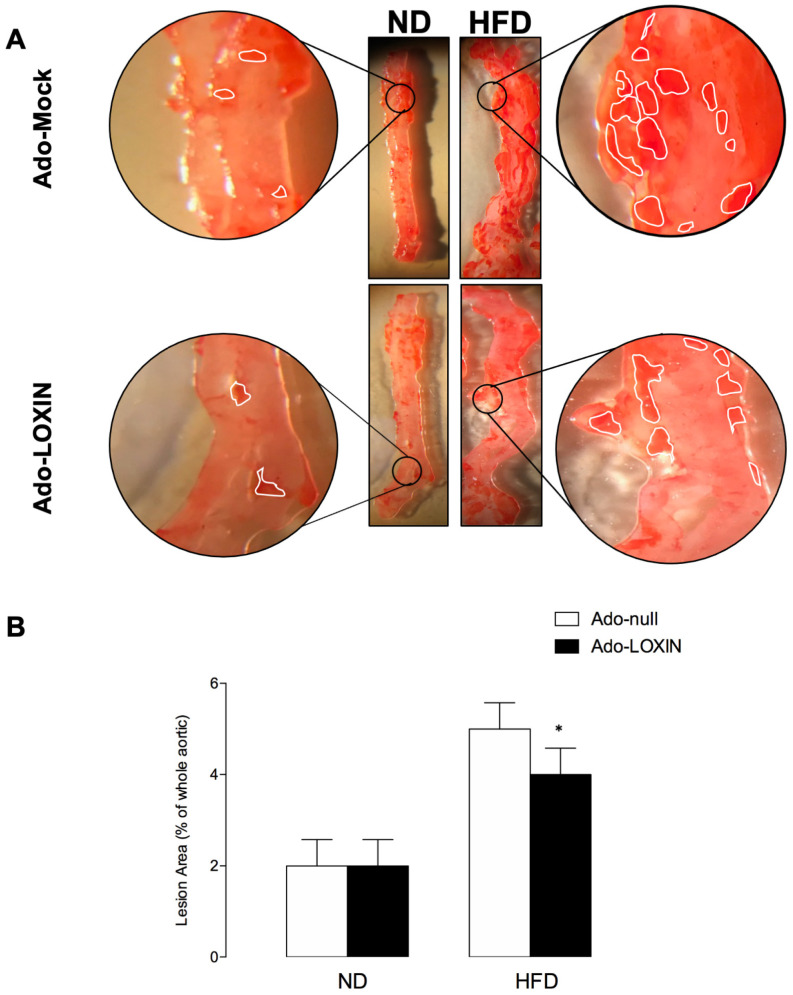
Effects of Ad-LOXIN on lipid deposition in aorta atherosclerosis in mice fed a high-fat diet. A total of 20 C57BL6 mice were divided into two groups, 10 normal diet (ND) and 10 high-fat diet (HFD) mice, and all specimens were fed for three months. Mice were divided into two groups and infected with an adenovirus vector (1 × 10^8^ Plaque-forming unit, pfu) expressing LOXIN (Ado-LOXIN) or empty vector (Ado-Null) via the tail vein. (**A**) Representative photographs of oil-red O-stained aortic plaques and pooled data of the effects of Ado-LOXIN on the percentages of plaque area are shown. The high magnification imaging was performed in the area delimited. (**B**) Quantification is the area occupied by lipid deposits (remarked) in the aorta. Data are expressed as the means ± SEM (*n* = 5). * *p* < 0.05, against the control.

**Figure 4 ijms-23-07329-f004:**
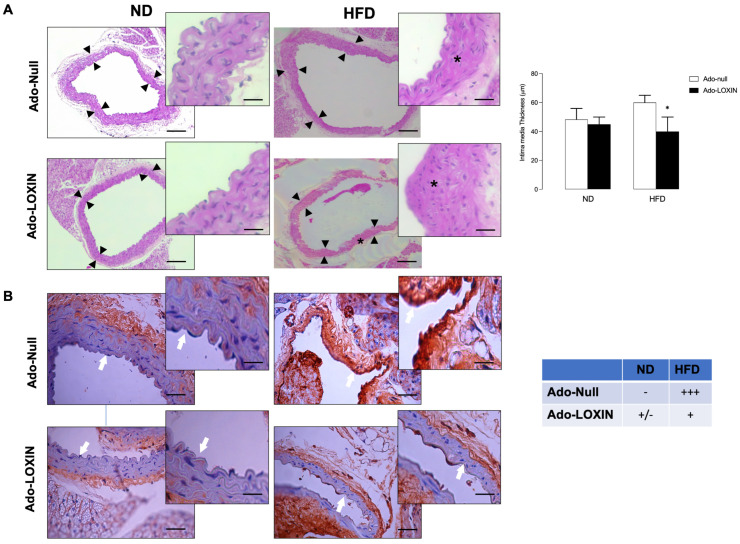
Effects of Ado-LOXIN on expression Lox-1 in mice fed a high-fat diet. (**A**) Representative microphotographs of hematoxylin-stained cross-sections of aortic plaques. * Represent perivascular alteration. Scale bars indicate 150 µm. The figure on the right shows the quantification of the aorta thickness (*n* = 5). (**B**) Immunohistochemical staining of antibody anti-Lox-1 was performed in permeabilized tissue sections obtained after Ado-LOXIN or empty vector injection, Ado-Null fed a normal diet (ND) or high-fat diet (HFD). Arrows indicated endothelial cells positive for Lox-1. Scale bars indicate 25 µm. The figure on the right shows the quantification of Lox-1 in crosses (*n* = 5). The high magnification imaging was performed in the area delimited in pictures A and B. Statistical significance is represented as * *p* < 0.05.

**Figure 5 ijms-23-07329-f005:**
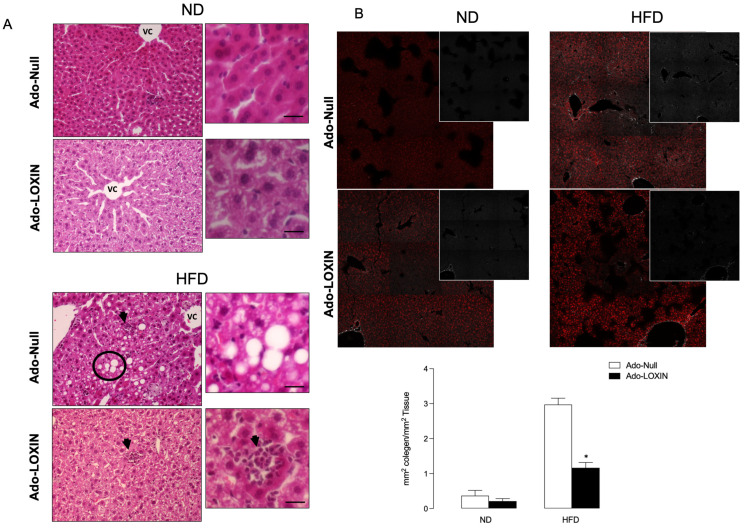
Histopathological analysis of liver sections. (**A**) Representative pictures of mice divided into two groups, normal diet (ND) and high-fat diet (HFD) mice and infected with an adenovirus vector expressing LOXIN (Ado-LOXIN) or empty vector (Ado-Null) via the tail vein. Arrows represent inflammatory cell infiltration, and the circles indicate hepatocellular hypertrophy or damaged hepatocytes. Vc, central vein. (**B**) Images of collagen fibers in liver tissue were recorded with second-harmonic generation (SHG) microscopy. Collagen deposits are shown in Sirius Red staining or grey (insert). Values are expressed as mean, and SEM *n* = 5. Scale bars indicate 10 µm. The figure above shows the quantification of collagen fibers (*n* = 5). Statistical significance is represented as * *p* < 0.05.

**Figure 6 ijms-23-07329-f006:**
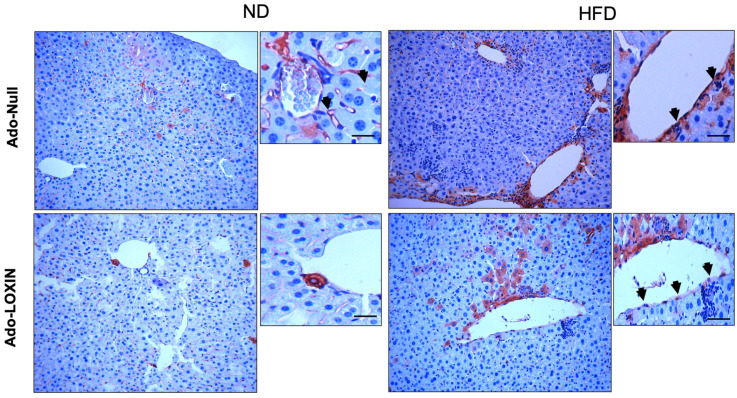
Characterization of Lox-1 in mice liver. Representative Lox-1 immunohistochemical staining for Lox-1 (original magnification, X400). The arrows indicate cells positive for Lox-1. Scale bars indicate 10 µm.

**Figure 7 ijms-23-07329-f007:**
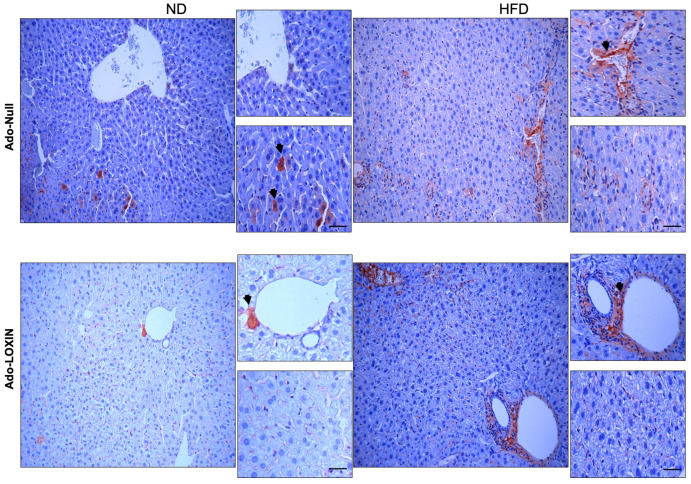
Hepatic TNF-α accumulation in the liver of each treated group of mice. Representative histological findings with immunohistochemical staining for TNF-α (original magnification, X400). The arrows indicate cells positive for TNF-α. Scale bars indicate 10 µm.

**Figure 8 ijms-23-07329-f008:**
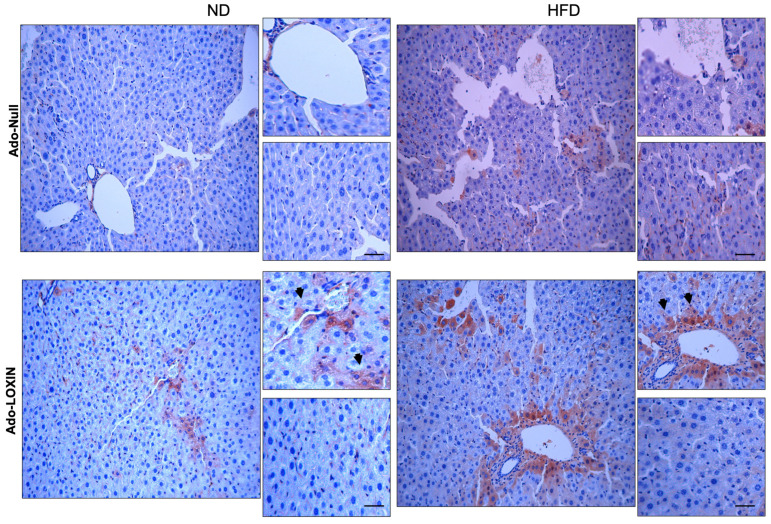
Hepatic IL-6 accumulation in the liver of each treated group of mice. Representative histological findings with immunohistochemical staining for IL-6 (original magnification, X400). The arrows indicate cells positive for IL-6. Scale bars indicate 10 µm.

## Data Availability

Data are available upon request. Contact Claudio Aguayo.

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
