# Peer review of "Loxin Reduced the Inflammatory Response in the Liver and the Aortic Fatty Streak Formation in Mice Fed with a High-Fat Diet"

_ijms, 2022, doi:10.3390/ijms23137329_

Round 1

Reviewer 1 Report

Summary:

The work of Reyes et al. investigates the role of LOXIN on inflammatory responses in the liver and aorta. The study is built upon previous results and adds on additional in vitro experiments to titrate the optimal concentration of oxLDL to investigate Lox-1 expression. The aim of this study is to investigate the role of LOXIN (a truncated protein of Lox-1) in vivo using C57Bl6 mice. They show on a histological level that viral overexpression of human LOXIN in mice has an effect on inflammatory parameters liver associated with down-regulation of Lox-1 in endothelial cells. The study is interesting but the presentation and results should be described more extensively. Also, the histological pictures should be quantified. To improve the impact of the manuscript and for a better understanding of it, some observations need to be addressed.

Major comments:

The Western blot on figure 1c shows bands at 30kD while the protein of Lox-1 is reported to be 50kD. The authors should elaborate on this. There is also no reference in the figure of which band is which protein. The authors need to verify that the band on the Western blot is actually the protein Lox-1, e.g. by using protein standards. 

The use of different cell types such as rat (H4-II-E-C3) and human (DU145) cells to show LOXIN overexpression reduces oxLDL internalization is not conclusive. The authors should use the proper controls in this experiment for each cell type with and without protein overexpression.

The authors need to show that viral treatment of overexpression of human LOXIN works in mice by analyzing for instance target genes downstream of Lox-1. A list of downstream effectors was published in 2021 in ATVB, an important paper that was not referred to in this manuscript. Barreto et al. (Role of LOX-1 as a cardiovascular risk predictor (ATVB. 2021;41:153–166))

C57BL6 mice have a predominant HDL lipoprotein profile and very low VLDL and LDL levels. Therefore it is not common to use C57BL6 mice to study atherosclerosis. Also because many other mouse strains to study atherosclerosis are available (see: doi:10.1161/ATVBAHA.107.142570). The authors should elaborate on this in the limitations of the study in the discussion section.

The Ado-LOXIN seems to have an effect on ALT and AST in the ND-fed mice (compared to Ado-Mock). Also, the Ado-LOXIN seems to have a protective effect on ALT and AST. The authors should elaborate on this effect and how to use this for the rest of the manuscript.

The text referring to Figure 2B is not supported by the figure. I cannot identify any changes in the endothelial structure nor on the lesions in the lumen of the aorta of HFD-fed mice. This needs to be adjusted and quantified at the group level.

The authors should elaborate on how to score ballooning of hepatocytes other than hypertrophic liver cells. Also, the authors need to quantify biochemically that there is less hepatic accumulation of lipids (determine liver lipids) and fibrosis (determine hepatic total collagen).

Minor comments: 

The resolution of figure 1 is very poor. Legends of the axes are not well readable. What does the asterisk stand for in table 1?

Author Response

.

Reviewer 2 Report

In this study the authors investigated effects of LOXIN, a truncated protein lacking the oxLDL binding domain in LOX-1, on oxLDL uptake in the cells and lipid accumulation in the aorta and the liver of mice fed with high-fat diet. The authors suggest that LOXIN overexpression inhibited LOX-1 mediated incorporation of oxLDL in hepatoma cell line. The authors also indicated that systemic overexpression of LOX-1 in mice suppressed high-fat-diet induced formation of aortic atherosclerosis and fatty streak and fibrosis in the liver. Even though the authors’ attempts might be potentially interesting, the current manuscript contains plenty of concerns that need to be improved to fully support the authors’ conclusion. Overall, the image data are of low quality and some of the data presented here are preliminary and do not fully support the authors’ proposed interpretation.

  • In abstract, the description in line 29-30, regarding interaction of LOXIN with LOX-1, is not fully supported by the current study. The additional experiments and results would be necessary.
  • LOX-1 and beta-actin should be indicated for each band in Fig 1C.
  • In particular, the image shown in Fig 1E is of poor quality and it is impossible to distinguish individual cells. In addition, the data do not sufficient to justify the conclusion that oxLDL internalization was reduced by LOXIN overexpression, due to lack of data for GFP-expressing control cells that would incorporate more oxLDL.
  • Fig 2 also contains images of low quality, especially Fig 2A, since each atherosclerotic lesion formed in the aorta is not clear.
  • The description in line 141-142 indicating that lymphocytes and polymorphonuclear leukocytes accumulate in the liver of high-fat-diet fed mice is not fully supported by the present data shown in Fig 3.
  • The area of lipid accumulation and fibrosis in the liver shown in Fig 3 should be quantified in combination with immunohistochemistry analysis of LOX-1 expression shown in Fig 4 to evaluate effect of LOXIN expression in mice.
  • Overall, the current manuscript is poorly articulated and difficult to read. Lots of description for data are not presented in a logical order and therefore are difficult to follow. Also, there are a lot of words containing hyphens. Should be reorganized completely.
  • Line 139-153 seems to be description of data for Fig 3, but not indicated.
  • The statement in Line 151-152, regarding infiltration of monocytes, macrophages and neutrophils, is not sustained by the results presented here.
  • The statement in Line 163-164, regarding LOX-1 expression in perivascular inflammatory cells, is not supported by the results presented here.

Author Response

.

Round 2

Reviewer 1 Report

The authors have adjusted the manuscript substantially and it did improve many parts of the manuscript. However, there are still some open issues related to the representation of the data, legends of the figures, and the statements in the text that need to be covered before publication.

The authors have improved the Western blot by adding the names, protein markers, and the correct molecular weights. However, the axes in figure 1 are still poor and not readable. The data in the Western blot is not representative. It shows a small increase in Lox-1 at 50ug/ml, while the figure shows a 2.5-fold increase in Lox-1/b-actin ratio. This cannot be attributed to lower b-actin. Also, the 100 ug/ml ox-LDL condition shows almost no Lox-1 protein, while the ratio in the figure is similar to 0 and 1 ug/ml conditions. This should be adjusted. 

The text added in lines 79-81 is very confusing. It is a summary of the paragraph above, not referred to any figure. It would help if Figure 1 is split into 2 figures if it is described over 2 paragraphs.

The authors have added figure 2, however, the order of the figures should be adjusted (Figure 3 is described before Figure 2). Also, the axis of ALT levels should be starting at zero. The authors need also to test the significance of ALT and AST in HFD-fed mice as compared to ND. If this is not significant, this should be mentioned in the text.

The authors did a very excellent job in better presenting the data of the atherosclerotic lesions in the aortic arch. However, the legend in figure 3 states n=10 per group, while the actual number is n=5 per group. I would prefer to show the standard deviation instead of  SEM. If SEM is used, the authors should at least check if the number is properly described. In this case, the number of samples is n=5.

The pictures in figure 4 are not from the same part of the aorta for the HE staining and the Lox-1 staining. Although otherwise stated in the text and the rebuttal, there are no aortic plaques visible in the HFD group. Similar structures in the vascular media are visible in both ND and HFD. The scale-bare for showing the scale is not in all pictures and is not mentioned in the legend. These issues need to be addressed. If the authors would like to make the statement “…reduced the lipid deposit in aortas and vessel intima disorganization” AND “we observed that Ado-LOXIN infection reduced the expression of Lox-1 compared to mice infected with Ad-Null “these data need to be quantified.

According to the representative pictures in figure 5B, the collagen levels are reduced in Ado-LOXIN mice of the HFD group compared to Ado-null infected animals. This is also quantified in the histogram. However, in the ND group collagen amount is increased in the Ado-LOXIN as compared to Ado-null mice. This needs to be addressed in the text and discussed, perhaps it links to ALT level differences?  The authors did a great job to include SHG collagen images, however, it would be of added value for the orientation of the reader to also show the HE-staining of the SHG-collagen images (or simple Sirius Red stainings). The authors should include in the legend whether the error bars in the histogram are expressed in Standard Deviation of SEM and also include the number of animals used for this analysis.

In the legend of Figure 7, the authors wrote “arrows indicate TNFa positive cells”. However, there is only one arrow visible in the figure where no staining is present. This does not seem to be correct.

Methods section: please indicate in which part of the aorta the sections were taken.

minor comments: Typos noticed

in line 70: live = liver

in line 71: Xhrs

line 91: concertation

table 1: NFD à HFD

line 479, colic acid à cholic acid

line 164, reference to figures should have a similar format, Fig. 5a à Figure 5A

The authors should check consistency in their writing. In the text, they state LOXIN, while in the figures it is often Loxin.

Line 263 à states “porque?” this should be clarified

Author Response

Dear reviewers,

We were pleased to be encouraged to send a revised version of our manuscript (IJMS-1705833) for your consideration.

We have conducted additional text revisions that the Reviewers suggested,

We thank the Reviewers for the constructive remarks, which helped improve the manuscript.

Dr Claudio Aguayo

Department of Clinical Biochemistry and Immunology

Faculty of Pharmacy

University of Concepción

Concepción, Chile

Phone: 56- 41-2207196

Reviewer 2 Report

-     Overall, the authors sincerely responded to some comments from the reviewer, and many points have been improved. However, I regret to say that Fig 1E is not yet completed due to lack of control experiments using a GFP-expressing adenovirus vector. In addition, the cell images seem to be curious since GFP expression in the cells is not clear and looks unnatural. The reviewer therefore cannot evaluate LOXIN overexpression-mediated reduction of oxLDL internalization on the basis of the current data set. If the authors cannot add the appropriate data, Fig 1E would be largely dispensable.

-       Table 1 and Fig 2 are duplicated.

-       Line 164-166: The immune cells indicated with arrow in Fig 5A cannot be identified as distinct cell types such as neutrophils/polymorphonuclear cells.  

Author Response

(The authors gave the same response as above.)

Round 3

Reviewer 1 Report

I would like to thank the authors for carefully replying to my comments in the rebuttal letter. 

Reviewer 2 Report

All the points raised in the previous version of the manuscript have been properly addressed by the authors. The reviewer sincerely appreciate the authors substantial efforts.